Escherichia coli transcription factors of unknown function: sequence features and possible evolutionary relationships

Duarte-Velázquez Isabel 1
de la Mora Javier 2
http://orcid.org/0000-0003-2780-5223 Ramírez-Prado Jorge Humberto 3
Aguillón-Bárcenas Alondra 1
Tornero-Gutiérrez Fátima 1
Cordero-Loreto Eugenia 1
Anaya-Velázquez Fernando 1
Páramo-Pérez Itzel 1
Rangel-Serrano Ángeles 1
Muñoz-Carranza Sergio Rodrigo 1
Romero-González Oscar Eduardo 1
Cardoso-Reyes Luis Rafael 1
Rodríguez-Ojeda Ricardo Alberto 1
http://orcid.org/0000-0001-6973-0595 Mora-Montes Héctor Manuel 1
Vargas-Maya Naurú Idalia 1
Padilla-Vaca Felipe 1 padillaf@ugto.mx
http://orcid.org/0000-0003-4332-3734 Franco Bernardo 1 bfranco@ugto.mx
1 Biology, División de Ciencias Naturales y Exactas, Universidad de Guanajuato , Guanajuato, Guanajuato , México
2 Departamento de Genética Molecular, Instituto de Fisiología Celular, Universidad Nacional Autonoma de Mexico , Mexico City , México
3 Unidad de Biotecnología, Centro de Investigación Científica de Yucatán, A. C. , Mérida, Yucatán , Mexico
Steczkiewicz Kamil
Electronic publication date: 2022 Jul 20
Publication date: 2022
Volume: 10
Electronic Location ID: e13772
Received 2022 Mar 29; Accepted 2022 Jul 1
Copyright: © 2022 Duarte-Velázquez et al.
Copyright year: 2022
Copyright holder: Duarte-Velázquez et al.
License: This is an open access article distributed under the terms of the Creative Commons Attribution License, which permits unrestricted use, distribution, reproduction and adaptation in any medium and for any purpose provided that it is properly attributed. For attribution, the original author(s), title, publication source (PeerJ) and either DOI or URL of the article must be cited.
License URL: https://creativecommons.org/licenses/by/4.0/

Keywords: Transcription factors of unknown function, Mobile elements, Sequence codon bias, Synteny, Escherichia coli, Structural features

Funding: CONACyT CB 182671, INFR-2013-205744, and CIBIOGEM-264456 This work was supported by CONACyT grants CB 182671, INFR-2013-205744, and CIBIOGEM-264456 grant. The funders had no role in study design, data collection and analysis, decision to publish, or preparation of the manuscript.

==============================
Organisms need mechanisms to perceive the environment and respond accordingly to environmental changes or the presence of hazards. Transcription factors (TFs) are required for cells to respond to the environment by controlling the expression of genes needed. Escherichia coli has been the model bacterium for many decades, and still, there are features embedded in its genome that remain unstudied. To date, 58 TFs remain poorly characterized, although their binding sites have been experimentally determined. This study showed that these TFs have sequence variation at the third codon position G+C content but maintain the same Codon Adaptation Index (CAI) trend as annotated functional transcription factors. Most of these transcription factors are in areas of the genome where abundant repetitive and mobile elements are present. Sequence divergence points to groups with distinctive sequence signatures but maintaining the same type of DNA binding domain. Finally, the analysis of the promoter sequences of the 58 TFs showed A+T rich regions that agree with the features of horizontally transferred genes. The findings reported here pave the way for future research of these TFs that may uncover their role as spare factors in case of lose-of-function mutations in core TFs and trace back their evolutionary history.

Introduction

The Escherichia coli genome was sequenced and published in 1997 (Blattner et al., 1997), describing the total coding capacity of this organism and positioning this bacterium as a model organism for prokaryotic cells. With the E. coli genome sequence available, thousands of published studies have assessed this organism’s physiology. The E. coli K-12 genome comprises 4.6 Mbp with a coding capacity of 4,501 genes, 4,284 protein-encoding genes, 89 tRNAs, 22 rRNAs, 58 regulatory RNAs, 36 other RNA molecules, 186 pseudogenes, and 1,264 genes of unknown function (Karp et al., 2018). Studying genes of unknown function, although complicated, renders an important step towards genome reduction by synthetic biology means in microbial organisms (Martínez-García & de Lorenzo, 2016).

Transcription factors (TFs) are the key elements for regulating spatiotemporal gene expression and adapting to environmental cues. These proteins are trans-acting factors that contain a DNA-binding domain and either a domain that senses a signal or allows them to interact with the RNA polymerase (Seshasayee, Sivaraman & Luscombe, 2011). TFs act either globally or locally. In the first case, the TF can regulate several genes, thus, several cellular functions, or in the latter case, a small number of genes (Seshasayee, Sivaraman & Luscombe, 2011). There are discrepancies in the total number of TFs present in E. coli, but thus far, 210 TFs are annotated, 58 are of unknown function (Karp et al., 2018). In the EcoCyc smart tables, there are 249 annotated TFs (http://regulondb.ccg.unam.mx/menu/download/computational_predictions/files/TF_predicted.txt) rendering a broad regulatory repertoire. Key questions regarding the TFs of unknown function are: Are they real TFs? DNA-binding experiments support this. Also, why some of the few characterized TFs of unknown function are differentially transcribed in various conditions? What could their evolutionary origin be?

In bacterial evolution, acquiring genomic island is a central topic, especially the transfer of pathogenicity-associated genes (Desvaux et al., 2020). One of the best-studied gene transfer examples is phage-mediated, which ultimately can lead to pandemic strains (Tassinari et al., 2020). Therefore, the study of genome dynamics takes a central spot in potentially pathogenic organisms.

Genomic islands contain special features, such as a lower G+C content, the presence of phage genes in the vicinity of virulence genes, and the presence of insertion sequences or repeated sequences. Genomic islands with particular functions such as metabolic, symbiotic, or the resistance to certain environmental insults and tRNA loci (da Silva Filho et al., 2018). But little attention has been focused on genes in the genome that may be related to novel functions acquired by horizontal transfer.

One example of this mechanism is the transcription factor GmrA, a horizontally acquired gene in E. coli O157:H7. This gene controls flagella synthesis by interacting with the promoter sequence of fliA and the sigma factor of the RNA polymerase (Yang et al., 2018). This suggests that some TF may be acquired horizontally, but the focus has been centered on their presence in pathogenicity islands.

Almost the majority of the TF in E. coli K-12 has been recently analyzed for their binding activity to DNA and the rest of the TFs of E. coli by gSELEX analysis (Ishihama, Shimada & Yamazaki, 2016). In some instances, the binding of TF plays an anti-silencing role, and in others depend on ligands for gene activation (Ishihama, Shimada & Yamazaki, 2016). Recently, the hierarchy of TF showed that the least minority of them bind to single targets, and a hierarchy has been proposed, from the minority as single target regulators, local regulators, global regulators, and nucleoid-associated regulators. Also, some of the TFs are located in prophages. Shimada et al. (2021) show that they may become active in regulating host genes, and host TFs may, in turn, also regulate prophage target genes. One key aspect that may support this is the estimation of the Codon Adaptation Index or CAI (Puigbò, Bravo & Garcia-Vallvé, 2008). The CAI value helps evaluate the impact on the preference for synonymous codons, how selection has affected the codon usage, and ultimately renders a sequence with a higher preference for translation. Strong biases in this value are accompanied by variation in G+C content, suggesting that certain genomic modules or genes may have been acquired horizontally as reported for viral genes adopting the codon usage of the host cell (Sharp & Li, 1987; Puigbò, Bravo & Garcia-Vallvé, 2008).

The features of TFs of unknown function have been poorly analyzed. Our goal is to analyze the sequence and structural features of TFs of unknown function to uncover the following: generate a hypothesis of their role and origin, address structural signatures for their physiological function, and datamining to assess their expression during different conditions.

In the present study, 58 TFs of unknown function have sequence features that stand out. Of the total of these TFs in E: coli, 14 are located in prophages, six of unknown function. They contain differences in sequence or structural elements that are conserved among other TFs of unknown function, not only constraint to the family previously identified in the genome annotation. The Codon Adaptation Index (CAI) of all 58 TFs analyzed showeds approximate values to those from annotated and functional TFs. Still, the G+C content in the third position of each codon varies greatly. Analysis of TFs of unknown function showed that they had predicted structural similarities, not confined exclusively to the DNA binding domain. Additionally, microarray data mining shows that all TF analyzed here show expression. Many of them correlate with STRING functional predictions and even show a putative network between them, suggesting a role in cell physiology.

Materials and Methods

Sequence retrieval

The overall workflow followed in this work is summarized in Fig. S1. Protein and nucleic acid sequences were retrieved from KEGG database (Kanehisa & Goto, 2000), using the accession numbers listed by Ishihama, Shimada & Yamazaki (2016). Also, Gao and colleagues have studied the TFs of unknown function DNA-binding sites (Gao et al., 2021) Here, focus on the TFs reported by Ishihama, Shimada & Yamazaki (2016) since they confirmed the DNA binding sites of these TFs throughout the genome by Systematic Evolution of Ligands by EXponential enrichment (SELEX), and the same group has carried subsequent work.

Sequence analysis

All 58 TFs were localized in the genome of E. coli. Highlighting the position of palindromic repeats, mobile elements, and prophage elements was done with CGView (Grant & Stothard, 2008).

Sequence comparison and analysis were carried out with Clustal Omega (Sievers et al., 2011) using five Iterations and other options in the default settings. The result was confirmed using MEGA.

Protein structure models

Protein structure prediction was conducted using AlphaFold2 (Jumper et al., 2021) with the default options, using the API hosted at Söding lab based on MMseqs2 server (Mirdita, Steinegger & Söding, 2019), generating five models per protein, using the most accurate for structural alignment (see below). Complementary reference structures were retrieved from the AlphaFold2 database at https://alphafold.ebi.ac.uk/, hosted by the European Bioinformatics Institute. These models are indicated in the figures with the corresponding accession number of this database. Model confirmation was done using the recently deposited structures of the E. coli proteome and using verify3D server (Bowie, Lüthy & Eisenberg, 1991; Lüthy, Bowie & Eisenberg, 1992). All models passed the quality control threshold. Reference annotated and functional TFs were downloaded from the AlphaFold2 database. In the case of the Xre-family, DicA was used for structural comparison (AlphaFold2 database accession AF-P06966-F1).

Protein models’ structural alignment was conducted with RaptorX (Källberg et al., 2012; Källberg et al., 2014) and mTM Align using the default options (Dong et al., 2018a, 2018b). Structural alignments showed a Root-mean-square deviation (RMSD) between 1.22 and 5.24. No relevant differences were observed with either platform. Visualization of protein models for obtaining individual images arranged as in structural alignments was conducted with PyMOL (Schrödinger & DeLano, 2020). A rainbow color scheme and cartoon representation were used to assess the localization of the N-terminal (blue) and C-terminal (red) ends.

Sequence motif discovery

For assessing conserved and potentially unique sequences, amino acid sequences were analyzed with GLAM2 using the default options (Frith et al., 2008). The best score predicted motif was mapped and identified in the protein sequences using MAST using the default options (Bailey & Gribskov, 1998). Manually, each sequence isindicated with the family type they are annotated.

For promoter sequence analysis for A+T rich sequence stretches, 400 bp sequences plus the ATG for each gene were downloaded from KEGG (Kanehisa & Goto, 2000). Then, the sequences were analyzed with GLAM2 using the default options. The highest score motif was mapped against all promoter sequences using MAST with the default options. Manually, each sequence is indicated with the family they are annotated. Also, the best score motif was scanned against known TFs binding sites using Tomtom with the default options (Gupta et al., 2007).

Additionally, promoter sequence prediction was conducted using two tools. One was NNPP using a cutoff of 0.8 (Reese, 2001). The second tool was PePPER (University of Groningen, Groningen, The Netherlands), a web server to predict prokaryote promoter elements and regulons (de Jong et al., 2012). The analysis with PePPER was conducted with the default options.

Sequence composition analysis

The nucleotide sequence of each TF, along with annotated and functional TFs were analyzed for G+C content at the third position of each codon. The expected Codon Adaptation Index (CAI) was estimated using the E-CAI webserver with the default options (at http://genomes.urv.es/CAIcal/E-CAI) (Puigbò, Bravo & Garcia-Vallvé, 2008). This tool renders a statistically significant CAI value by comparing the G+C content against the normalized CAI value calculated by the quotient of the CAI of a gene and its expected value. This tool provides a better statistical analysis to find whether a gene is biased only to mutation or a true codon usage adaptation. The data were plotted using Microsoft Excel by plotting only the %G+C at the third position vs. the normalized CAI value or ordered from the lowest to the highest CAI value and plotted separately. As a comparison standard, the same parameters were estimated for 58 annotated and functional TFs, i.e., those with known functions.

Gene location and putative interacting partners

Synteny and gene context was analyzed for each TF using GeConT 3 webserver with the default options (three neighboring genes upstream and downstream for each query) (Abdala, Ciria & Merino, 2008). Each hit was confirmed by Blast analysis with the default options and non-redundant sequence search (Zhang & Madden, 1997), for assessing true homology of each hit in bacteria different from E. coli. Sequences showing insertions, different sequences, or structural variations were retrieved and further analyzed.

Interacting or functional partners for each TF were assessed with STRING with the default options (von Mering et al., 2005; Franceschini et al., 2013). A complete list for each TF is provided in File S3, containing the details of the co-occurrence or co-expressing partners for each TF. Also, the full list of the TFs analyzed in the present work were analyzed by STRING to discover potential functional relationships. It was analyzed by STRING to find potential functional relationships among them.

Gene expression data mining

The 58 TFs analyzed were screened in the GenExpDB online webserver hosted by Oklahoma State University (at https://genexpdb.okstate.edu/databases/genexpdb/) to browse the data of 216 data sets containing 1,292 microarray experiments of E. coli. Since published datasets were analyzed differently and the GenExpDB contains normalized data, the data cannot be re-analyzed from raw data, or re-scaled, the datasets from different studies cannot be compared quantitatively, only qualitatively with the provided data. Thus, we selected the data from microarrays that contain contrasting experimental conditions, such as the expression of a small RNA involved in the iron response, three strains adapted to high temperature (41.5 °C) and a time course of different stressing conditions, for small RNA (RyhB) iron response (Massé, Vanderpool & Gottesman, 2005); the expression analysis of three temperature adapted strains (Riehle et al., 2003); and the analysis of four stressing conditions using a time course (Jozefczuk et al., 2010). The data was collected from these microarrays, including the data for LysR, AraC, and GntR for comparison. Data were analyzed by heatmapper (Babicki et al., 2016). The data collected from each condition for each TF analyzed here was plotted using the average linkage clustering method and Spearman Rank Correlation for distance measurement. Also, from GenExpDB other examples were analyzed. The 58 TFs analyzed here that showed either up or down-regulation were also considered also a,correlation with STRING predictions. Only some examples are shown, but many TFs showed regulation in various conditions. The data is shown in Fig. S7 and in File S2 the reference for each experiment and the detail of each experimental condition where regulation was observed in each microarray experiment.

Results

The nature of a TF is determined by the DNA-binding domain and the presence of a regulatory domain (Iyer & Aravind, 2012) that links the TF to a particular cellular function, ligand, or response to an environmental cue (Browning, Butala & Busby, 2019). In Fig. 1, we show a schematic representation of the relevant features of TFs. In Fig. 1A we show that TFs activity depends on two main domains, a DNA-binding module, and a ligand or regulatory module. The outcome of the TF activity may depend on ligands, covalent modifications such as phosphorylation, the binding of another protein partner or the subcellular location, therefore resulting in the repression or the activation of target genes. In Fig. 1B the prototypical structure of a TF is shown. PurR, a GalR/LacI familyGalR/LacI family member, contains a canonical HTH DNA-binding domain and a regulatory domain that, upon binding of the effector (hypoxanthine and guanine), allows the conformational change that permits DNA binding (Schumacher et al., 1994).

Figure 1 Transcription factors in bacteria are proteins mainly with two domains.

In (A), TFs activity depends on the location or accessibility to the target cis sequence, the binding of a partner protein, and the binding of ligands or covalent modifications such as phosphorylation. Once activated, most TFs dimerize and bind to target sequences in the vicinity of the core promoter sequence that either repress or activate transcription. In (B), an example of a typical TF shows the DNA-binding domain and the regulatory domain, in this case, a ligand-binding domain for hypoxanthine. The protein shown here is PurR, a LacI-family member (PDB accession number 2PUB) (Schumacher et al., 1994).

Previously, Iyer & Aravind (2012) reported that the number of TFs correlates with the genome size, Posing a more complicated task in determining the relevance and function of many TFs. The number of TFs of unknown function in E. coli requires further study to assess if they are real TFs truly functional. There are 58 TFs in this category for which binding sites have been determined experimentally (by SELEX) and that are scattered throughout the genome (Ishihama, Shimada & Yamazaki, 2016).

TFs of unknown function sequence analysis

The sequence of the 58 TFs of unknown function reported by Ishihama, Shimada & Yamazaki (2016) were selected, and their sequence features were analyzed based on similarity and structure.

Figure 2A shows that TFs of unknown function are split into 13 distinctive groups in the phylogenetic tree. Ancestral nodes can be identified for some of the groups (indicated with red arrows), suggesting that no strong bias is observed between family members. While some clustering is observed for the rest of the groups, this is less evident than the observed in the first clade. This suggests that even though TFs of unknown function are annotated to particular families, more divergence than expected can be found between them.

Figure 2 Sequence and structural features of 58 TFs of unknown function.

When comparing all transcription factors, homology is focused on the DNA binding domain. To prevent bias, sequence comparison was carried out using a guide tree. After five iterations using Clustal Omega, we identified clusters of proteins unrelated to them and clusters of closely related protein sequences. Identified ancestral nodes are indicated with a red arrow. AlphaFold2 models were used for structural comparison to find homology beyond the DNA binding domains using RaptorX (Källberg et al., 2012, 2014) to facilitate common structural cores. Names are placed just beside the protein on each alignment and indicated with an arrow. Refer to Fig. S1 for each protein predicted structure in the rotation as shown in the alignment for easier comparison. In the case of groups 1, 2, and 3, *indicates an overall comparison with the most outlier protein (YgfI) is presented in Fig. S2.

Next, a structural comparison analysis was conducted to assess whether the groups had limited structural conservation or if divergence was significant than those observed in standard sequence comparison. The structural alignments were performed among the members of each group of proteins from the phylogenetic analysis (Fig. 2B).

The alignments revealed that the proteins are diverse structurally. In more distant TFs, diversification was more evident. Each alignment indicates only the conserved structure on each set (Fig. 2) and can be traced to the representative TF that assigned the family for each TF of unknown function, as shown in Fig. S2. In Fig. S3, the comparisons of the three first groups are provided, including the outlier YgfI, and the three distinct positions of the N-terminal region can be identified (DNA-binding domain).

The rest of the TFs show lower sequence and structural similarity even in small groups such as the 12 and 13 groups, which should show closer homology between them. These results collectively suggest that TFs of unknown function may be less prone to retain characteristics from an ancestor and accumulate more variability. The structural features indicate that the domain or domains involved in regulating the TF activity may diverge more frequently than the DNA-binding domain, thus retaining the features that allow classification to a specific family type.

In addition to e the actual variation observed in the structural comparisons, and a structural comparison against each classification’s representative annotated and functional TF family member was conducted. In Figs. 3A and 3B, the structural alignment of all TFs was performed by each set of branches shown in Fig. 1.

Figure 3 Structural features of TFs of unknown function contain strong similarities with bona fide TFs that determine the family classification.

(A) Structural alignments using mTM Align (Dong et al., 2018a, 2018b) show each transcription factor’s color code. The name color indicates the TF on the alignment. Plot (B) has the same alignments shown in (A) but highlights the common core in magenta on each set. Arrow indicates that the alignment was rotated to allow visualization of the common core. Name and AlphaFold2 database accession numbers indicate reference TFs for each family.

In Fig. 3A, structural comparisons indicate evident similarity, especially against with the TF that designates each family. All groups except for Xre, showed better structural similarity than in Fig. 2 (RMSD values range from 1.22 to 4.5). Also, groups such as LysR, RutR, and AraC, show more diverse spatial configurations than the parental TF (RMSD values range from 1.22 to 2.1). For example, LysR TFs have three distinct conformations for the N-terminal domain. In AraC-family, YijO contains an N-terminal domain that is more divergent than the rest of the TFs of this group, suggesting a sensory role for this domain in this TF.

When comparing the common core on each set, the similarity of each protein alignment is lower in the first sets and higher in the more distant sets, as shown in Fig. 3B. This result suggests that the TFs closely related to their parental TF have higher diversification than those less divergent to their parental TF.

All 58 TFs were compared and analyzed for motif discovery using GLAM2 and then searched for the best scoring motif using MAST to assess the possibility of sequence or sequences that may be further conserved. In Figs. S4A (motif) and S4B (position), only one best matching motif was found in some TF sequences, mostly near the N-terminal domain where the DNA-binding domain is located. One exception, YgfI, is located near the C-terminal domain and far from the DNA-binding domain (Fig. S4C). Belonging to the LysR family the motif is found in ten TFs [YbhD (set 1), YnfL (set 1), YhaJ (set 2), YeeY (set 2), YdhB (set 2), YafC (set 3), YcaN (set 3), YhjC (set 3), YgfI (between sets 3–4), YbdO (set 8)]. Belonging to the GntR-family, three TFs were found [YdcR (set 6), YegW (set 6), YihL (set 6-GntR)]. Three belong to DeoR-family [YjiR (set 6), YgbI (set 5), YihW (set 5)]. Two belong to RipR-family [YtfH (set 12), YfhH (set11). And one for LacI-family [YcjW (set 13)], NagC-family [YphH (set 7)] and IclR-family [YagI, (set 10-IclR)].

The representative motif suggests that it is constrained to the TFs of unknown function that belong to the most abundant families of transcription factors but is absent in the rest of the TFs analyzed here. This feature may be related to other factors such as genome position, horizontal gene transfer, or differences in sequence features that may regulate the translation of each gene.

Sequence analysis for HGT traits

To test the idea that the coding sequence of each TF may be evolutionary divergent, %G+C content in the third position of each codon and the normalized Codon Adaptation Index (CAI) was calculated for TFs annotated and functional, and those of unknown function. This analysis may allow finding whether a gene is biased only to mutation or if a faithful codon usage adaptation exists; if so, some of the genes encoding for TFs of unknown function may be on the verge of becoming functional or be fully functional.

In Fig. 4A, the analysis of %G+C content against normalized CAI showed clustering of annotated and functional TFs (indicated with dashed lines, showing some outliers of both kinds). Those of unknown function showed a more scattered profile and tended to bear a lower %G+C content in the third position of each codon. This suggests an overall variation in the %G+C content. Also, this result suggested that these TFs may have a more variable CAI, thus having repercussions on their translation.

Figure 4 %G+C content and normalized CAI suggest a bias in TFs of unknown function.

(A) Comparison of %G+C at the third position against the normalized CAI values of 58 TFs of unknown function (blue dots) and annotated and functional TFs (orange dots) as described in the Methods section. Dashed lines were included to indicate the major cluster between annotated and functional TFs and those with unknown functions, excluding two outliers of annotated and functional TFs. With the clustering observed for each dataset, in (B), the normalized CAI was ordered from the lowest to the highest value (purple data points) and then plotted along with the %G+C content (red data points), indicating each TF. Horizontal dashed lines were used to indicate the limits of normalized CAI values for annotated and functional TFs to facilitate comparison with the TFs of unknown function. In the case of TFs of unknown function, the family that each one belongs is shown. The vertical dashed line indicates the separation of TFs of unknown function from those with known regulatory roles.

To further test this, CAI values were ordered from the lowest to the highest in annotated and functional transcription factors and in the 58 TFs of unknown function. Then, simultaneously plotted the %G+C variation at the third position. In Fig. 4B, the variation in CAI values is in the same range for both types of TFs, contrary to the assumption made by observing the clustering pattern. Still, the %G+C values differ significantly among TFs of unknown function (indicated by a dashed line for comparison between the two groups). TFs variations are in the range of 46.4–69.5%, while in the TFs of unknown function vary between 22.1–83.3%. This was further verified using 15 LysR experimentally characterized TF (Fig. S5), showing the same trend as the overall TFs shown in Fig. 4. In annotated and functional.

Overall, this suggests that most of the LysR-family of TFs of unknown function may show lower translational efficiency and accumulate more mutations, leading to a lower %G+C content than the genome’s overall %G+C content. Also, those having higher normalized CAI values may potentially be gaining functional roles in E. coli and are worth further in vivo analysis.

TFs promoter sequences show potential “promoter islands”

Previous reports indicate that regions enriched with A+T pairs may be responsible for integrating foreign genes into the recipient bacterial regulatory network (Daubin & Ochman, 2004). Also, the increased frequency of promoter-like sequences has been recognized as a possible signature of HGT acquired genes (Huang et al., 2012).

Recently, experimental evidence of such events has been reported (Bykov et al., 2020) and corroborated the hypothesis that several promoter sequences hinder RNA polymerases’ activity by suppressing the expression of foreign genes. Also, the regions of HGT genes result in accumulating A+T-rich regions and creating promoter-like elements.

To test this possibility, sequence motif discovery was conducted to prevent any bias of real promoter sequences, using 400 bp upstream regions of all 58 TFs, including the ATG sequence.

In Fig. S6, using GLAM2, we discovered two motifs or stretches rich in A+T that are located in the 400 bp upstream of the ATG (panels A and C). The position of these motifs and the family that each TF belongs to is shown in panels B and D. These results suggest that most of the sequences analyzed had at least one predicted promoter.

For ycaN no promoter was found with NNPP, while PePPER found a sequence that may function as a promoter in the negative strand. For ydaW, yfjR, and yidL no evident promoter sequence was found. The rest contain at least one predicted promoter sequence, while most of the promoter sequences containing the motif shown in Fig. S6A contains more than one promoter sequence. The second motif (panel B) showed less in the TFs analyzed, most had more than one predicted promoter sequence. Thus, the exceptions are yihL, yahB, yphH, and ycnE, which lacks a precise promoter sequence. For motif 2 (panel B), yfjR and ycaN lack a predicted promoter sequence, and yphH and yneL contain only one promoter sequence.

Overall, the analysis shown in Fig. S6 suggests that some of the TFs of unknown function may have been either acquired by HGT or are the result of gene duplication.

Expression patterns for the 58 TFs

To address if a role can be predicted for the TFs and their conservation across species, synteny, STRING analysis, and microarray data mining were conducted.

First, we selected three microarray datasets that contain contrasting situations, the expression of RyhB regulatory RNA, the effect of selective pressure at 42 °C (comparison between adapted vs. ancestral strains), and a time course for different stresses (Massé, Vanderpool & Gottesman, 2005; Riehle et al., 2003; Jozefczuk et al., 2010).

In Fig. 5, a heatmap analysis of the expression data retrieved from GenExpDB shows that clustering is observed between TFs of unknown function. From the data, we note the following:

Figure 5 Transcriptional datamining of the 58 TFs of unknown function.

Heatmap of three microarray data covering the effect of RhyB expression and iron induction, E. coli adapted strains to 41.5 °C, and four different stressing conditions ranging cold, heat, oxidative and metabolic stress. Heatmap includes the clustering of the data using average linkage and Spearman Rank Correlation. For each experiment, the condition used is indicated at the bottom. Dashed lines separate each dataset. The family for each TF is shown in the color code displayed on the right. Black arrows indicate the position of three annotated and functional TFs (LysR, GntR and AraC).

First, for the effect of expressing the small RNA RhyB or the conditions tested in this experiment, it is noted that most of the Tfs are repressed with some exceptions such as YbhD and YgeK. In other instances, FeSO4 induces their expression such is the case of YcaN, YeiL, and YjiR.

Second, in the temperature-adapted strains, strong expression in one or either of the three adapted strains is observed for the comparison between the temperature-adapted strain vs. the ancestral strain. Such is the case for: YafC, YgbI, YhjC, YihL, YiiF, and YieP. For YegW, two temperature-adapted strains showed expression, but one showed repression. In other instances, strong repression is observed, such is the case for: LysR (included here as an external reference), YbaO (in one of the three strains), YbeF (in the three strains) and YiaU. This suggests that some of the TFs may perform as either buffering TFs during strong selective pressures or have a low expression profile that is activated during selection.

Finally, the analysis of cold, heat, oxidative, and metabolic stress, using a time series of experimental data, can lead to finding clusters of TFs that may show differential expression. As shown in Fig. 5, sets of TFs are found with opposing expression patterns, either for cold or heat stress for example. A trend can be found for some of the TFs analyzed here but not as evident as found for cold or heat stress with oxidative and metabolic stress. This suggests that these TFs form an underlying network of buffering TFs that may become active depending on the selective pressure.

In GenExpDB there is a compendium of 216 microarray data; in many instances, we found specific experimental setups that showed strong regulation (either positive or negative), suggesting that the TFs analyzed here have other regulatory conditions and, in some instances, correlate with the STRING prediction.

The analysis of all microarray data is summarized in Fig. S7, where predicted synteny, if present, is indicated for each TF. STRING relevant hits are indicated, and snapshots of the microarray data indicating the reference and condition corresponding to each experiment are included. In File S2, the reference for each microarray experiment, the description of the microarray experiment, and the relevant expression data for the heatmap are provided.

The data shown in Fig. S7 regarding synteny is further detailed in File S3, where information about the neighborhood in the genome of each gene and the flanking genes or synteny registered in other organisms is provided. Also, in this file, STRING hits are provided, emphasizing in each case if the hit is related to co-occurrence, co-expression, or both. In most cases, the hits are irrelevant, a decisive conclusion cannot be drawn. The data is summarized in Fig. S7.

The relevant observations drawn from the analysis shown in Fig. S7 are as follows:

First, YgeK and YneL are annotated as pseudogenes. Figure S8 shows snapshots of heat maps from 216-microarray data (GenExpDB), indicating that YgeK and YneL are expressed and regulated in many conditions.

Second, YneL showed an interesting synteny arrangement (Fig. S6) that was restricted to E. coli strains. In Fig. S9, a snapshot of the genetic context of this gene shows that this gene has two different domains. The gene has an extra domain annotated as a P-pilus assembly protein and another where it is either annotated as a pseudogene or contains an AraC-type DNA binding domain. Also, the flanking genes are different in several strains. This feature was further analyzed, and AlphaFold2 modeling revealed that the BL21 YneL indeed contains a domain strikingly like FimA. The models from DH10B strain and MG1655 strains are similar but not identical. The AraC-type DNA binding domain is located at the N-terminal portion of the protein. It is conserved when comparing the YnfL model from MG1655 and BL21 strains (Fig. S10, structural alignment). Also, in the W strain, the sequence shows low similarity in the C-terminal region, which once modeled shows a remarkable difference (Fig. S10), suggesting that this TF may differ significantly in E. coli strains others than the ones analyzed here. Similarly, YdaW also shows important structural variation prediction regarding differences in sequence length in different strains, for instance, the sequence in MG1655 vs. O55:H7 (Fig. S11). This suggests inter-strain variation that may ultimately result in divergence.

Third, although little correlation was found between the STRING hits and the expression data, there are cases where evident regulation is observed.

Fourth, TFs belonging to LysR-family have fewer expression data than the TFs from the more distant groups, except for Xre-family TFs, which showed more defined roles. The case of YiaG shows expression under different conditions, clear expression during stress and during stringent response. In Fig. S8, a snapshot of all 216 experiments where this TF has expression or repression under different metabolic conditions, RNA decay, and growth conditions.

Lastly, some TFs were found to have fewer microarray data that supported that they were expressed in a specific condition, except in different mutant backgrounds and in an adaptative condition such as UV irradiated or temperature stressed adapted strains, suggesting that in laboratory adaptation experiments, these TFs may become active by accumulating mutations and thus becoming ‘rescue’ TFs (see for example YneJ and reference [2] in File S2 and Fig. S6).

In the case of YfjR, no predicted promoter sequences are found. Nevertheless, strong expression is observed in microarray datasets (Fig. S6). This DeoR transcription regulator shows consistent activation during biofilm formation (Herzberg et al., 2006). A mutant lacking this TF showed a 65% reduction in biofilm formation, suggesting that this TF belongs to the group of TFs and effectors for biofilm formation. For instance, YfjR, a CP4-57 prophage (location and information at https://ecocyc.org/gene?orgid=ECOLI&id=G6685#tab=TU) derived TF may have belonged to a set of spare TFs. The other TFs analyzed in this work that are located in prophages are: YdaW (https://ecocyc.org/gene?orgid=ECOLI&id=G7369#tab=TU), YbcM (https://ecocyc.org/gene?orgid=ECOLI&id=G6302#tab=TU), YjhJ (https://ecocyc.org/gene?orgid=ECOLI&id=G7913#tab=TU), YagA (https://ecocyc.org/gene?orgid=ECOLI&id=EG12338#tab=TU), and YagI (https://ecocyc.org/gene?orgid=ECOLI&id=G6144#tab=TU). One special case is YgeK, which is located near prophage derived genes (https://ecocyc.org/gene?orgid=ECOLI&id=G7475#tab=TU).

Overall, the data collected in Fig. S7 and File S3, contains information for further testing the phenotype of individual mutants of each TF analyzed here. Also, compelling data suggests that the expressed genes in microarray and the promoter sequences data suggest that these TFs may be buffering when losing annotated and functional TFs.

Genomic location of TFs

The analysis shown here of the 58 TFs, suggests a putative duplication or HGT origin. Analyzing the genome elements near the TFs, several repeated sequences or mobile elements may explain the bias in codon usage, differences in %G+C content in the third position, and the presence of A+T rich sequences or an excessive number of predicted promoters. In Fig. 6 and Fig. S12, the genomic location of each TF is shown, with the comparison to the genomic map where mobile elements and repeated sequences are located. Transcription factors of the LysR-family tend to be clustered, while the rest show less relevant clustering.

Figure 6 The genomic landscape of the 58 TF analyzed.

In (A), G+C content and skew are shown, along with cryptic prophage and mobile elements in the complete genome. Only the TFs are shown in (B), indicating the family that each TF belongs to. ▴ Indicates the approximate regions of high transcription rate and ▴ indicates the lowest transcription regions according to Scholz et al. (2019). The list on the right of the figure suggests the highest to the lowest value for active regions (▴), and the repressed regions (▴) indicate the lowest to the highest of the repressed areas.

By comparing the data presented by Scholz et al. (2019) related to the highest and lowest transcriptional active regions of the E. coli genome, we found that most of the LysR-family TFs are located in low expression regions of the genome. The GntR-family TFs are situated in the highest transcriptional active region. Partially, YiaG is located between a low and a high activity region. This observation correlates with the diminished transcriptional activity found in the 216-microarray data analyzed. Also, as shown in Fig. 5, some of the LysR-family TFs found to be active in artificially selected strains, such as temperature-stressed adapted strains, become active under stressing conditions. Noteworthy, they are located in less active genome regions (see Fig. 6).

The following genes that are present in these active regions showed high transcriptional activity in microarray datasets: YafC (temperature adaptation and isobutanol usage), YjhJ (activated in glucose to lactose or acetate to lactose shift), YtfH (activated in heat stress), YidC (activated in heat stress), YijO (biofilm formation and transcription termination), YihW (heat stress and glucose to lactose shift), YihL (temperature adaptation, reference 2 in File S2, also motif 1 (Fig. S6A) in promoter plus a single promoter sequence), YiiF (transcription termination), YieP (temperature adaptation, reference 2 in File S2 and cold stress), YidP (stress response and metabolism, also motif 1 (Fig. 5A) in promoter sequence and four predicted promoters), YidL (biofilm formation). The silent regions are YiaU, which is only active in transcription termination analysis using bicyclomycin and shows strong activation in a genome-reduced strain (MDS42, reference 12 in File S2). YhaJ is also silent in temperature adaptation strains (reference 2 in File S2) and metabolism during glucose to acetate shift (reference 6 in File S2).

The only exception regarding genome localization near highly transcribed regions is YjiR (region II in Fig. 6B). YjiR showed repression in Wt and DksA deficient cells. Also, YjiR is only active in cells treated with serine hydroxamate to generate amino acid starvation (see reference 17 in File S2). It also contains motif 1 (Fig. S6A) and three promoter sequences.

Only a couple of examples of these TFs were found to possess either A+T motifs described in Fig. S6, suggesting that the positions in which these genes belong are related to the active or silent regions of the genome.

The genomic location that is transcriptionally active (Scholz et al., 2019), also shows the lowest content of mobile or repeated elements (Fig. 6 and Fig. S12).

The genomic map also suggests that independently of the position and GC skew, the normalized CAI values and %G+C content in the third position may be independent of the localization of the genome, supporting the hypothesis that these genes may be duplication or HGT events. Finally, the TFs located in the higher transcriptional activity region of the genome may be relevant examples for further experimental analysis.

Discussion

An enormous effort has been conducted to achieve a complete image of the regulatory network of transcription factors in the model organism E. coli. The complex regulatory network comprises seven σ factors and ~300 predicted TFs (Ishihama, 2012). 270 TFs have been purified and analyzed by systematic SELEX, using a careful and direct approach, through the generation of a genomic library of small DNA fragments and purified proteins to discover binding sites (Ishihama, 2012; Shimada, Ogasawara & Ishihama, 2018a). With this approach, several targets have been identified and paved the way for assessing the role of TFs in cell physiology and environmental responses. Obvious limitations include knowing the exact combination of TFs and σ factors at a given moment or the lack of certain TFs if others may render certain plasticity to bypass the regulatory voids.

The complement for this impressive achievement is the Keio collection, a set of single in-frame deletion mutants that can serve as a platform for further assessing the role of each TF (Yamamoto et al., 2009; Baba et al., 2006), with the need of obvious controls and careful analysis due to polar effects (Mateus et al., 2021).

For example, the screening of such mutant collection uncovered the role of novel genes involved in regulating swimming and swarming motility in E. coli (Inoue et al., 2007) or the discovery of fluorophores that can be accumulated in the cytoplasm depending on the deficiency of transporters (Salcedo-Sora et al., 2021).

The study of TFs or proteins of unknown characteristics can uncover novel functions, such as the recently found role for PlaR (previously named YiaJ). PlaR main role is to allow E. coli to tribe outside warm-blooded organisms, survive with plant-derived nutrients, and form biofilms by regulating a limited but sufficient set of genes (Shimada et al., 2019).

The evolutionary aspect of TFs of unknown function

The evolution of the bacterial genome is a hot topic. Mainly because of the strong relationship between the movement of pathogenesis-related genes and the origin of virulence traits and thus, rendering E. coli as highly adapted pathogen to a myriad of conditions and niches (Desvaux et al., 2020). Desvaux et al. (2020) described genomic integration sites near tRNA genes, resulting in the presence of direct repeats. This is the case for many regions in the genome where TFs are located, showing ‘scars’ of repeated sequences in the neighboring genes or the presence of tRNA genes (File S3).

The surveillance of these events is relevant not only for epidemic purposes, like the one reported by Tassinari et al. (2020), where a mobile gene (sopE) which acts as a guanine nucleotide exchange factor that activates Rho-GTPases Cdc42 and Rac-1 in the host cells, causing membrane ruffling. is a mobile genetic element. It is present in half of the species of S. enterica subspecies I.

In the analysis shown here, the TF was found in some strains of E. coli annotated as YneL. Still, three showed the inserted FimA homologous domain, which encodes for the P-pilus assembly protein, pilin FimA. The analysis showed that YneL contained several configurations according to the deposited sequence. The only study where the mutant for YneL coding sequence reports no variation in resistance to ultrasound (Spiteri et al., 2017). However, the authors do not discuss the reason behind using the mutant lacking this gene or the mechanism that is derived from the resistance to this condition.

The synteny analysis and the location of each gene revealed that many possess different lengths, and most showed high synteny conservation (for example, those belonging to LysR-family) to low conservation, as found for the lower groups in Fig. 2.

The only TFs studied here that have known targets are: YdcI (Total regulated genes: 2, total regulated operons: 1 and total of binding sites: 1, total regulatory interactions: 1); YeiL (Total regulated genes: 1, total regulated operons: 1 and total of binding sites:1, total regulatory interactions: 1), and YhaJ (Total regulated genes: 1, total regulated operons: 1 and total of binding sites:1, total regulatory interactions: 1) (RegulonDB, Santos-Zavaleta et al., 2019). Perhaps using a less stringent analysis, other weak targets may arise relevant to their function. This also correlates with the low expression rates found in Fig. 5.

TFs structural analysis

Overall, the PDB contains 789 E. coli experimentally determined protein structures, but with the implementation of AlphaFold2 and AlphaFold2 database (Jumper et al., 2021; David et al., 2021), the assessment of structural features in unknown proteins is faster and highly accurate and provides a starting point for future experimental approaches. Also, the careful and human-monitored analysis should be taken to avoid misinterpreting the predicted structure, especially in disordered regions (Ruff & Pappu, 2021).

In Figs. 2 and 3, the similarity of each TF is constrained to the DNA-binding domain (N-terminal end), but once the structure is compared, we found structural features that are highly conserved but shows differences in certain domains compared to the annotated and functional family member TF that was used to assign the family membership. This explains why in some instances, synteny is found in highly divergent organisms.

The structural analysis agrees that low normalized CAI sequences mainly belong to LysR-family TFs and show a higher variation in %G+C content in the third position; suggesting either HGT genes or duplicated genes exhibiting independent variation. Strikingly, the active TFs showing defined functions may be associated with a more stable genomic location showing different normalized CAI values but retain the %G+C on the third position as for core genes.

TFs of unknown function are transcribed

The results regarding expression showed that the TFs of unknown function are transcribed in a myriad of conditions. Those located in the lagging-strand tend to be more expensive energetically speaking, consistent with the model that shows that active genes are located in the leading strand. Those located in the lagging strand tend to be more expensive, energetically speaking, resulting in mutational bias (Gao et al., 2017). It also correlates with the highly expressed genes found in the areas of the genome with peaks of high expression vs. low expression sites that are flanked by prophage or repeated sequences or mobile elements (Scholz et al., 2019). Interestingly, the GLAM2 signature found in some TFs correlates with the position in the genome of low transcriptional activity except for YihL (GntR), YjiR (DeoR), YihW (DeoR) and YtfH (RipR) that are located in the highly transcribed region.

Perhaps many of the silent copies of TFs analyzed in the present study may be silenced by H-NS due to the presence of A+T sequences in the promoter sequence (Lang et al., 2007).

The findings presented here show that the genes encode TFs with variable sequence and structural features correlate with the observation that HGT genes show low selective pressure. These genes are expressed more in poor environments and are susceptible to transcription-driven mutagenesis, depending on the growing conditions (Feugeas et al., 2016).

TF landscape

Transcription factors vary greatly depending on the target sequences, from single targets (Shimada, Ogasawara & Ishihama, 2018b) many and diverse binding sites and targets depending on the nature of the bacterial organism (Flores-Bautista et al., 2020). Also, bacterial genomes show diversity in all aspects, such as size, G+C content, gene distribution, among others. For instance, in E. coli, the smallest genome belongs to the BL21 strain (Lukjancenko, Wassenaar & Ussery, 2010). This points towards the need to design analyses to assess the micro diversity in bacterial genomes (Touzain et al., 2010), focusing on locus related to high mobility such as tRNAs and small non-coding RNAs (Germon et al., 2007; Sridhar & Rafi, 2007).

The landscape of the TFs is also related to the essentiality of the genes. Finding new conditions where any gene may be needed, the comparison of several strains is needed. This demonstrates that the essentiality correlates with the epistatic effect of mobile elements; resulting in the activation of genes that may not be essential under laboratory conditions but are needed in the environment (Rousset et al., 2021). In the future, the analysis of mutants lacking any of these TFs in strains different from the MG1655 may render important information of the role of the TFs analyzed here, as described for GmrA that in the O157:H7 strain controls the expression of flagellar genes (Yang et al., 2018).

STRING analysis with the 58 TFs studied here showed a network that contains mainly TFs with fewer members in the overall TF families in E. coli. In Fig. S13, the network comprises mainly TFs that are less conserved and belong to AraC, RutR, and, as expected, the LysR-family due to its vast number of members. The consistency in the expression data found in the time course analysis during cell stress (Fig. 5) correlates with YgfI, YfeD, YfeC, YiaU, YfiE, and YddM, which seems to be functionally linked, as shown in Fig. S13. Overall, the data in Fig. S13 suggests that the TFs of unknown function may constitute a buffering regulatory network that may be functional during adaptation or when annotated and functional TFs are missing or have deleterious mutations.

The observations suggest that using careful analysis, human monitored and manually curated data (Méndez-Cruz et al., 2020), allows the discovery of novel features in proteins that often are neglected due to the difficulty of designing experimental designs approaches to discover gene function.

The analysis of TFs of unknown function has revealed interesting and novel features; such as the case of YieP, which recently was demonstrated that this TF binds to intergenic regions far from the regulatory regions. Lower global binding or binding in regions of genes with putative functions has rendered an interesting model for further research of novel regulatory modes (Gao et al., 2018). Also, the most common binding site for YieP is downstream of the RNA polymerase binding region, rendering an out-of-the-ordinary TF (Gao et al., 2018).

With all the above, here we propose three scenarios for the origin of these TFs:

First, the CAI and %G+C suggest that many of the TFs of unknown function were acquired horizontally but a long time ago. This hypothesis is supported when phylogenetic reconstructions are done using the same 58 annotated and functional TFs. This is further reinforced when a pairwise analysis is carried out and compared both sets of TFs (Fig. 7). This indicates that both kinds of TFs are divergent, mainly because of differences in the regulatory and DNA binding domains, which provide specificity to the ligands and DNA sequences.

Figure 7 Global sequence comparison between the 58 TFs analyzed as annotated and functional or canonical TFs against the 58 TFs of unknown function.

Protein sequences were aligned using MUSCLE and then analyzed in Aligmentviewer. Pairwise identity 2D map was generated for each set; sequence order is given in File S4. In (A), a pairwise comparison between the 58 canonical TFs is provided. Clustering is observed for each family. In (B), the comparison between the 58 TFs of unknown function is provided. In (C) overall comparison between TFs of unknown function against TFs of known function. Scale indicates at 0 that 0% identity is found, and 1 indicates 100% identity.

Second, if the transfer did occur, it may have been from bacteria phylogenetically close to E. coli. This is supported by the synteny analysis, where most neighboring genes belong to closely related bacteria, although adjacent genes are conserved in closely related bacteria, and exceptions are observed.

Third, the challenging idea is that most of the 58 TFs of unknown function originated by duplication and no selective pressures forced selected them to be functional. This is supported when comparing the expression patterns found in Fig. 5. Most of the TFs of unknown functions showed a clustering expression pattern under different conditions in this analysis. Perhaps, the strains used in the lab have been so domesticated that the TFs of unknown function have lost their ability to be expressed, or the activation signal is something only found in the usual ecological niche that E. coli thrives. This suggests that annotated and functional transcription factors may mask their activity. Also, the homology between the two kinds of TFs is low (Fig. 7).

Further research is needed to address each TF studied here and its contribution to E. coli cell physiology. In turn, this will shed lighton how effective genome reduction can be achieved and whether any of these TFs are required as quasi-essential.

To address this, we now have a complete set of data sets with 17,000 genome-wide binding maps that have provided integrated information to define functional modules and pathways (Baumgart et al., 2021), discovering new functional roles of TFs and an increase in DNA binding motifs in E. coli, that ultimately may serve as a powerful tool for future research of the transcriptional landscape of this organism. We cannot rule out a novel effect of metabolites that are now known that similarly drives genome structural rearrangements as heterochromatin, mediated by Hfq (Beaufay et al., 2021), that can ultimately displace TFs or generate structural impediments. This is relevant in the case of the TFs analyzed here since the Hfq binding site is enriched in A+T-rich DNA regions. Also, based on the recent finding that polyphosphate regulates Hfq, increasing prophage and transposon mobilization occurs in the absence of this polyanion (Beaufay et al., 2021).

During the review of this manuscript, a recent paper described the role of YiaU (now termed CsuR), that is required to produce membrane proteins of unknown function but is involved in biofilm formation, complement and antibiotics sensitivity (Shimada et al., 2022).

Supplemental Information

Supplemental Information 1 Workflow used in this work.

Aspects to be explored in the sequence and structural features of TFs of unknown function are indicated and the tools used.

Click here for additional data file.

Supplemental Information 2 Ribbon models of all the proteins are analyzed in Figures 2 and 3.

Each model is shown by group and in the approximate position shown in the structural alignments. The blue color indicates the N-terminal end, and the red indicates the C-terminal end. For groups 7 and 8, proteins are shown in the position for the overall alignment (Figure 2). The predicted family for each TF is indicated in red, and models with red frames indicate the reference structure for the representative member of the stated family of TF.

Click here for additional data file.

Supplemental Information 3 Structural alignment of groups 1, 2, and 3, including YgfI.

The relative position of each group is in relation to the most distant set of the phylogenetic comparison. YgfL is structurally similar to Group 1 in Figure 2, even though the sequence is more divergent.

Click here for additional data file.

Supplemental Information 4 Motif discovery in the 58 TFs amino acid sequences from E. coli.

Using GLAM2 (Frith et al., 2008) a motif was found with the following regular expression: L[ST][FI].E[AL]A[ER].LGG?[IV]SS[AT][LV]SR.[IL]?QKL?E..[GL] (Panel A). Then, using MAST (Bailey & Gribskov, 1998), the motif was scanned and mapped in the protein sequences, finding a positive consensus in the following sequences: YtfH (set 12-RipR), YhaJ (set 2-LysR), YafC (set 3-LysR), YdcR (set 6-GntR), YegW (set 6-GntR), YcaN (set 3-LysR), YbhD (set 1-LysR), YhjC (set 3-LysR), YjiR (set 6-DeoR), YgbI (set 5-DeoR), YeeY (set 2-LysR), YcjW (set 13-LacI), YphH (set 7-NagC), YihW (set 5-DeoR), YfhH (set 12-RipR), YnfL (set 1-LysR), YgfI (set 3–4-LysR), YihL (set 6-GntR), YagI, (set 10-IclR), YbdO (set 8-LysR), YdhB (set 2-LysR). Low hit sequences were also indicated. On each hit using MAST, the family of each TF is indicated using the color code indicated on top of the MAST result. For comparison with the localization of the motif, Panel C provides the Prosite database scan (Sigrist et al., 2012) of each TF to locate the DNA binding domain or other well conserved domains.

Click here for additional data file.

Supplemental Information 5 Motif discovery in the 58 TFs amino acid sequences from E. coli.

Using GLAM2 (Frith et al., 2008) a motif was found with the following regular expression: L[ST][FI].E[AL]A[ER].LGG?[IV]SS[AT][LV]SR.[IL]?QKL?E..[GL] (Panel A). Then, using MAST (Bailey & Gribskov, 1998), the motif was scanned and mapped in the protein sequences, finding a positive consensus in the following sequences: YtfH (set 12-RipR), YhaJ (set 2-LysR), YafC (set 3-LysR), YdcR (set 6-GntR), YegW (set 6-GntR), YcaN (set 3-LysR), YbhD (set 1-LysR), YhjC (set 3-LysR), YjiR (set 6-DeoR), YgbI (set 5-DeoR), YeeY (set 2-LysR), YcjW (set 13-LacI), YphH (set 7-NagC), YihW (set 5-DeoR), YfhH (set 12-RipR), YnfL (set 1-LysR), YgfI (set 3–4-LysR), YihL (set 6-GntR), YagI, (set 10-IclR), YbdO (set 8-LysR), YdhB (set 2-LysR). Low hit sequences were also indicated. On each hit using MAST, the family of each TF is indicated using the color code indicated on top of the MAST result. For comparison with the localization of the motif, Panel C provides the Prosite database scan (Sigrist et al., 2012) of each TF to locate the DNA binding domain or other well conserved domains.

Click here for additional data file.

Supplemental Information 6 %G+C content and normalized CAI suggest a bias in TFs of unknown function.

Here, 15 experimentally validated LysR family TFs were analyzed for the normalized CAI, ordered from the lowest to the highest value (purple data points), and then plotted along with the %G+C content (red data points), indicating each TF. Horizontal dashed lines were used to indicate the limits of normalized CAI values for annotated and functional TFs to facilitate comparison

Click here for additional data file.

Supplemental Information 7 Promoter sequence analysis of the 58 TFs of unknown function for sequence motifs.

In Panel A and C, Weblogo analysis derived from GLAM2 analysis of 400 bp and the ATG sequence for each TF of unknown function For panel A, the regular expression is [AT][ATG][ACG][AT][AT]TTTT.{0,75}[AT][AT]AA[AT][AG]ATG[ACT][ATG][AT]TT and for Panel B the regular expression is [ACT][TG][TG][AT]A[AT][AC]AT?[AC]AT[CG]A[ACT][AT][AT][AT]. Each sequence was then analyzed using MAST for relative position and abundance. Panels B and C show the map, the number of motifs shown on Panels A and C, and the number of predicted promoter sequences described in the Methods section. Low match results were also included (absence of red boxes). The family for each TF is indicated in the color code between panels B and D. Promoter prediction was conducted with NNPP, except for those sequences where no predicted result was found, was further analyzed with PePPER.

Click here for additional data file.

Supplemental Information 8 Data mining for the 58 TFs of unknown function.

The data presented is Synteny (using GeConT 3), putative interacting partners that may represent a function by STRING analysis and expression data from https://genexpdb.okstate.edu/databases/genexpdb/. Legend: Synteny* Presence of other arrangements in diverse bacteria. # Indicates that different arrangements are found in E. coli strains, although no synteny outside E. coli is found. STRING** Predicted functional partners or the relevant function of partners. Expression*** snapshot of data heatmap of representative experiments either relevant to STRING results or showing consistent regulation, and threshold expression are indicated in the far right of the figure. The reference for each experiment is shown here is indicated in brackets. On top of some datasets is displayed the corresponding condition that either expression or repression is observed. Data from different studies cannot be compared quantitatively, only qualitatively. All data details and expression profiles are presented in Files S2 and S3.

Click here for additional data file.

Supplemental Information 9 Snapshot of the overall transcriptional profile of three TFs.

Two (YgeK and YneL) pseudogenes and a highly regulated TF (YiaG) are shown. Data retrieved from GenExpDB (https://genexpdb.okstate.edu/databases/genexpdb/). Heat map corresponds to −3 to 3 thresholds.

Click here for additional data file.

Supplemental Information 10 Sequence analysis of YneL in different E. coli strains.

Sequence comparison was conducted with Clustal Omega, and synteny was determined with GeConT 3 tools. Dashed arrows indicate models of each case. The blue color indicates the N-terminal end, and the red indicates the C-terminal end. Synteny legend: blue box indicates AraC-type DNA-binding domain. The Orange box indicates P-pilus assembly protein, pilin FimA. FimA AlphaFold model (Accession number AF-P04128-F1) is shown.

Click here for additional data file.

Supplemental Information 11 Structural comparison between different YneL sequences, modeled with AlphaFold2.

Color code indicates the common overlap between structures, and magenta indicates the common core in each comparison. Strong sequence and structural conservation are found between FimA and YneL from BL21 strain.

Click here for additional data file.

Supplemental Information 12 Sequence comparison between YdaW sequences using Clustal W and AlphaFold2 models.

In strain O55:H7 a larger protein with a different structure is found. Orange box, MarR-type of DNA-binding protein. Light purple box indicates a sequence of an uncharacterized protein conserved in archaea, corresponding to the putative DNA-binding domain in YdaW.

Click here for additional data file.

Supplemental Information 13 Genomic map.

Genomic map for all 58 TFs showing simultaneously all the elements that suggest HGT: mobile elements, prophage elements, and repeated sequences.

Click here for additional data file.

Supplemental Information 14 String analysis of the 58 TFs analyzed here.

Each TF is manually indicated the family it belongs; for reference, on the left is shown the phylogenetic comparison shown in Figure 2. Additional regulators are shown with black arrows with its description.

Click here for additional data file.

Supplemental Information 15 Predicted promoter sequences in all TFs analyzed in the present work.

Predictions were conducted with NNPP or PePPER.

Click here for additional data file.

Supplemental Information 16 Additional references and data details for Figure S7 indicating the relevant data for each microarray analysis.

The TF analyzed is in bold.

Click here for additional data file.

Supplemental Information 17 A table containing all the details for synteny and STRING analysis for Figure S7.

Click here for additional data file.

Supplemental Information 18 Contains all PDB files, sequences and raw data for STRING, expression and CAI analysis used in this study.

Click here for additional data file.

Additional Information and Declarations

Competing Interests

Author Contributions

Data Availability

Bernardo Franco is an Academic Editor for PeerJ. Héctor Manuel Mora-Montes is an Academic Editor for PeerJ.

Isabel Duarte-Velázquez performed the experiments, prepared figures and/or tables, and approved the final draft.

Javier de la Mora performed the experiments, analyzed the data, prepared figures and/or tables, authored or reviewed drafts of the article, and approved the final draft.

Jorge Humberto Ramírez-Prado performed the experiments, analyzed the data, prepared figures and/or tables, authored or reviewed drafts of the article, and approved the final draft.

Alondra Aguillón-Bárcenas performed the experiments, prepared figures and/or tables, and approved the final draft.

Fátima Tornero-Gutiérrez performed the experiments, prepared figures and/or tables, and approved the final draft.

Eugenia Cordero-Loreto performed the experiments, prepared figures and/or tables, and approved the final draft.

Fernando Anaya-Velázquez performed the experiments, prepared figures and/or tables, and approved the final draft.

Itzel Páramo-Pérez performed the experiments, prepared figures and/or tables, and approved the final draft.

Ángeles Rangel-Serrano performed the experiments, prepared figures and/or tables, and approved the final draft.

Sergio Rodrigo Muñoz-Carranza performed the experiments, prepared figures and/or tables, and approved the final draft.

Oscar Eduardo Romero-González performed the experiments, prepared figures and/or tables, and approved the final draft.

Luis Rafael Cardoso-Reyes performed the experiments, prepared figures and/or tables, and approved the final draft.

Ricardo Alberto Rodríguez-Ojeda performed the experiments, prepared figures and/or tables, and approved the final draft.

Héctor Manuel Mora-Montes performed the experiments, analyzed the data, prepared figures and/or tables, authored or reviewed drafts of the article, and approved the final draft.

Naurú Idalia Vargas-Maya conceived and designed the experiments, analyzed the data, prepared figures and/or tables, authored or reviewed drafts of the article, and approved the final draft.

Felipe Padilla-Vaca conceived and designed the experiments, analyzed the data, prepared figures and/or tables, authored or reviewed drafts of the article, and approved the final draft.

Bernardo Franco conceived and designed the experiments, analyzed the data, prepared figures and/or tables, authored or reviewed drafts of the article, and approved the final draft.

The following information was supplied regarding data availability:

The raw data is available in the Supplemental Files.

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
