# Peer review of "Escherichia coli transcription factors of unknown function: sequence features and possible evolutionary relationships"

_PeerJ, doi:10.7717/peerj.13772_

## Round 0.1 · original submission · Major Revisions

The Reviewers were optimistic about the manuscript however, they raised a few concerns. Please respond to those carefully. Especially, consider applying the approach to some already experimentally studied TFs for validation as suggested by dr Monteiro as well as discuss the modeling part according to dr Knizewski's comments.

Generally, I feel the manuscript's language might be improved and the overall flow simplified. I'd also suggest working out image readability (at least for some), e.g. fig 4 which is unreadable. Maybe consider moving some of them into supplementary data?

Reviewer 1 ·

Basic reporting

Duerte-Velazquez and colleagues have presents analysis of 58 putative transcription factors. These TFs are known to be expressed genes that have been previously experimentally reported to bind to DNA.
From a technical perspective, the manuscript is well understood, and professional English is used throughout. The literature references are mostly adequate. The figures and tables are well-structured and sufficient raw data is provided.
The key points of their analysis are:
1. These genes are located at genome regions containing significant number of mobile genetic elements (MGE) and have promoter characteristics of duplicated or horizontally transferred genes (HTG). They display highly variable sequence/genetic landscape, which is consistent with organism need for highly variable regulation factors.
2. Yet, despite high variation, these genes display adaptation to expression in E.coli indicated mainly by codon adaptation similar to other known E.coli TFs.
3. These putative TFs have divergent protein sequence retaining 3d structure, in respective structural groups, that goes beyond DNA binding structural elements.
4. Some of these putative TFs show a correlation with some stress expression profiles, hinting on single process regulation.
Altogether, Authors speculate that these might be the features of backup TFs, or silent genes with TF evolutionary potential.

Experimental design

Authors have used the wide variety of data sources and methods to describe this putative TFs. The methods are sound and have been (with minor notes) described with sufficient detail. The research was sufficiently defined and is within the aims and scope of the journal.

Validity of the findings

The arguments and analysis results are sound, definitely valuable and intriguing. Although, proving backup TFs function (knock-outs) in the most cases will be experimentally challenging.

Additional comments

The reviewer has only minor questions concerning the manuscript:

1. In the abstract and further in the manuscript, the authors refer to various TFs as bona fide TFs, core TF. Please, clarify which TFs are these? Experimentally confirmed, annotated, etc? Is there a list of this core TFs?
2. Line 67: provided link leads to page showing error: ":ALL-PROTEINS-5 not found". Please provide query to reproduce said 248 TFs list.
3. Line 121: Isihama and colleagues in paper from 201 refers to 285 DNA-binding TFs that are to be analysed via SELEX screening (page 2070, last paragraph). Have this progress been finally reported by Ishihama and has it been included in the analysis?
4. Line 132 vs Figure 2: Authors cite publication on AlphaFold2 while claiming to use AlphoFold. Please clarify which method was used. This typo is also present in later parts of the manuscript (i.e. line 555).
5. Line 141: Authors have used two services to produce structural alignments. Was there a step to agree results of these servers? How were the differences resolved? Also, Figure 2 description name only one structural analysis server.
6. Line 149: GLAM2 was used to discover sequence motifs. In general GLAM2 can generate own alignments, how did this differ to structure-derived ones?
7. Line 159: Again, authors have used to tools to predict promote regions. What was the final result, an interjection of these results, a sum?
8. Line 234: Authors have used different tool to again compare and overlap structures (mTM). Did this results agree with Raptor results.
9. Line 558: When reviewing structural models and model overlaps, reader should have known which regions do not overlap due to structure divergence but are of high confidence, versus parts of models that are of low confidence. So that false hypotheses are not formed on poorly modelled regions. Can model confidence be included into supplementary information (unless column 11 of raw data models in Raw Data\PDB files correspond to confidence scores)?
10. Line 106: Which 14 of 58 analysed TFs are located in prophages. Please provide database link or some sort of evidence.
11. Line 106: These 14 TFs display sequence/structure divergence conserved with other putative TFs. Which TFs are these, are they also in prophages, or the presence in such regions does not correlate with conservation?
12. Line 109: How does the third position variation look like in "bona fide" TFs in comparison to analysed 58 TFs. Can you comment on the possible difference?
13. Line 132-140: I am confused which structures were finally used for further structural overlap and analysis, these made with AlphaFold (if so which of the 5 generated model), these downloaded from EBI or these from PDB?
14. Line 136: What was the process of model confirmation? Did the authors compare their AlphaFold models against experimentally determined structures of the same TFs?
15. Paragraph Results/TF of unknown function sequence analysis: The reviewer would love to see a figure not only with structural overlap of structures in groups but also domain annotation on sequence level (at least similar to motif locations in Figure 4), that would help to identify common domain context within clusters and easily compare then between groups.
16. Paragraph Results/TF of unknown function sequence analysis: have authors tried to compare structures from different sequence-based groups (aside 1-3)? Were there any similarities beyond DNA binding domain?
17. Line 283: Have the motif been mapped to domain context and to the 3d structure? Does this motif correlate with genome location, putative horizontal transfer origin hypothesis?

As a general comment, this reviewer considers Verify3D as an obsolete MQAP tools. There are better tools, however in this case, AlphaFold2 provides some confidence scores itself.

Reviewer 2 ·

Basic reporting

The topic addressed by the authors is very important and I see a lot of potential for the field. However, due to the complexity of the topic I recommend that authors consider rewriting some parts of the main text in order to make the reading more fluid.
English revision is mandatory.

Experimental design

The authors did not develop any new pipeline of platforms. Here the authors are exploring 58 unknown TF based on structural similarities, sequence homologies, and alignments. So, I am not convinced that this work is innovative but the description of available tools in the literature.

I struggle to understand what the goals were.

A Workflow for the work would be helpful.

Validity of the findings

As mentioned before I am not convinced that this work is innovative.
Before analyzing unknown TFs and their TF binding sites, the authors should validate the proposed method, for a group of experimentally validated TF in which the functions are well described.

Additional comments

The paper is too long and the main text is confusing and prolix.
The quality of the image resolution needs to be improved.

---

## Round 0.2 · accepted · Accept

The Reviewers accepted the revised version of the manuscript. Congratulations!

Reviewer 1 ·

Basic reporting

All my concerns and question have been addressed. I find them quite satisfactory.

1. Minor numbering differences between responses and actual manuscript did not hamper the re-review.

2. The case of YgeK gene being in vicinity of prophage-derived genes is not obvious to reviewer. Reviewer has found in quoted source genes from type III secretion system (Pat.Isl.) and some genes/domains from transposition process. The reviewer lacks the required expertise to validate prophage-derived origin of these genes.

I would recommend PeerJ to accept this article.

Experimental design

The experiment design did not change, and is OK. Even better, as some technical methods were additionally explained.

Validity of the findings

Authors have cross-checked tool results and formed hypotheses that require heinous experimental work to confirm. No additional comment on re-review in this section.